# Metformin Impedes Oxidation of LDL In Vitro

**DOI:** 10.3390/pharmaceutics15082111

**Published:** 2023-08-09

**Authors:** Christine Rossmann, Cornelia Ranz, Gerd Kager, Gerhard Ledinski, Martin Koestenberger, Willibald Wonisch, Thomas Wagner, Sebastian P. Schwaminger, Bruno Di Geronimo, Andelko Hrzenjak, Seth Hallstöm, Gilbert Reibnegger, Gerhard Cvirn, Margret Paar

**Affiliations:** 1Division of Medicinal Chemistry, Otto Loewi Research Centre, Medical University of Graz, 8010 Graz, Austria; christine.rossmann@medunigraz.at (C.R.); caennsi@gmail.com (C.R.); gerd.kager@medunigraz.at (G.K.); gerhard.ledinski@medunigraz.at (G.L.); willibald.wonisch@medunigraz.at (W.W.); sebastian.schwaminger@medunigraz.at (S.P.S.); bruno.digeronimo@medunigraz.at (B.D.G.); seth.hallstroem@medunigraz.at (S.H.); gilbert.reibnegger@medunigraz.at (G.R.); margret.paar@medunigraz.at (M.P.); 2Department of Pediatrics and Adolescent Medicine, Division of General Pediatrics, Medical University of Graz, 8010 Graz, Austria; martin.koestenberger@medunigraz.at; 3Department of Blood Group Serology and Transfusion Medicine, Medical University of Graz, 8010 Graz, Austria; thomas.wagner@medunigraz.at; 4BioTechMed Graz, 8010 Graz, Austria; 5Division of Pulmonology, Department of Internal Medicine, Medical University of Graz, 8010 Graz, Austria; andelko.hrzenjak@medunigraz.at; 6Division of Biomedical Research and Translational Medicine, Medical University of Vienna, 1090 Vienna, Austria

**Keywords:** antioxidants, atherosclerosis, diabetes, copper ions, low density lipoprotein, lipid oxidation

## Abstract

Metformin is the most commonly prescribed glucose-lowering drug for the treatment of type 2 diabetes. The aim of this study was to investigate whether metformin is capable of impeding the oxidation of LDL, a crucial step in the development of endothelial dysfunction and atherosclerosis. LDL was oxidized by addition of CuCl_2_ in the presence of increasing concentrations of metformin. The extent of LDL oxidation was assessed by measuring lipid hydroperoxide and malondialdehyde concentrations, relative electrophoretic mobilities, and oxidation-specific immune epitopes. Cytotoxicity of oxLDL in the vascular endothelial cell line EA.hy926 was assessed using the alamarBlue viability test. Quantum chemical calculations were performed to determine free energies of reactions between metformin and radicals typical for lipid oxidation. Metformin concentration-dependently impeded the formation of lipid hydroperoxides, malondialdehyde, and oxidation-specific immune epitopes when oxidation of LDL was initiated by addition of Cu^2+^. The cytotoxicity of oxLDL was reduced when it was obtained under increasing concentrations of metformin. The quantum chemical calculations revealed that only the reaction of metformin with hydroxyl radicals is exergonic, whereas the reactions with hydroperoxyl radicals or superoxide radical anions are endergonic. Metformin, beside its glucose-lowering effect, might be a suitable agent to impede the development of atherosclerosis and associated CVD. This is due to its capability to impede LDL oxidation, most likely by scavenging hydroxyl radicals.

## 1. Introduction

Metformin is a guanidine derivative and was initially extracted from the plant Galega officinalis (French lilac). It has been used as a glucose-lowering medication in humans for more than 60 years. Metformin is highly cost-effective and has proven to be safe. Metformin is considered as an insulin-sensitizer by virtue of its capability to reduce insulin resistance and decrease plasma fasting insulin levels [1,2,3]. In numerous clinical trials, metformin has been shown to decrease the blood glucose level by decreasing the hepatic glucose output as well as by inducing greater peripheral glucose uptake [4].

Metformin plays an important role in pharmacologic therapy for adults with type 2 diabetes. The American Diabetes Association recommends that metformin should be continued upon initiation of insulin therapy (unless contraindicated or not tolerated) for ongoing glycemic and metabolic benefit [5]. Metformin is effective, safe, inexpensive, and available in an immediate-release form for twice-daily dosing or in an extended-release form that can be given once daily. Because type 2 diabetes is a progressive disease, combination therapy might be required. Stepwise addition of medications to metformin is recommended to maintain glucose levels at target. More extensive initial combination therapy for more rapid attainment of glycemic goals and later combination therapy for longer durability of glycemic effect are required [5,6,7,8].

The principal side effects of metformin use are gastrointestinal intolerance due to bloating, abdominal discomfort, diarrhea, and lactic acidosis. Metformin may be safely used in people with reduced estimated glomerular filtration rates [5]. Moreover, metformin use could be associated with vitamin B12 deficiency and worsening of neuropathy [9]. These side effects occur very rarely.

Several additional beneficial properties have been attributed to metformin. For example, metformin affects the aging process [10] and exerts anti-inflammatory as well as anticancerous effects [11,12]. Moreover, metformin is capable of minimizing the elevated risk of cardiovascular disease in diabetic patients [13]. The capability of metformin to reduce atherosclerosis progression in individuals with type 1 diabetes has been confirmed by carotid artery intima–media thickness measurements, a surrogate outcome for atherosclerotic cardiovascular disease [2,14,15,16,17].

However, the mechanisms of action through which metformin exerts its antiatherosclerotic effects are not fully understood. Low-density lipoprotein (LDL), not atherogenic in its native state (nLDL), might become subject to oxidation in the subendothelial space of arteries by mechanisms involving free radicals and/or lipoxygenases.

LDL is the most important cholesterol-carrying particle in the circulation. The main role of LDL is to deliver cholesterol to both peripheral and liver cells. The interaction of dietary and genetic factors determines the amount of LDL cholesterol in the plasma. The liver generates very low-density lipoprotein, which is metabolized by lipoprotein lipase to intermediate-density lipoprotein. Hepatic triglyceride lipase converts intermediate-density lipoprotein to LDL, which is cleared from the circulation via the LDL receptors expressed in the liver and other cells [18]. Total cholesterol exceeding 200 mg/dL and LDL cholesterol exceeding 130 mg/dL are considered abnormal. In overweight people, lifestyle modifications are crucial (exercise and/or diet control) to lose weight. To decrease LDL levels, 3-hydroxy-3-methylglutaryl-coenzym-A (HMG Coa) reductase inhibitors are used. These drugs are capable of inhibiting the conversion of HMG Coa to the cholesterol precursor mevalonate [19]. Proprotein convertase subtilisin/kexin type 9 (PCSK 9) inhibitors also significantly decrease serum LDL levels by dampening the degradation of LDL receptors [20].

LDL lipids undergo peroxidation. Subsequently, certain aldehydes generated during lipid peroxidation modify the apolipoprotein B100 (apoB100), the protein part of LDL. The resulting oxidized form of LDL (oxLDL) is readily internalized by macrophages through a so-called “scavenger receptor” pathway. The macrophages change to foam cells, initiating processes resulting in atherosclerosis [21]. The liver assembles triglyceride-rich lipoproteins and secretes them into the circulation. Lipoprotein lipases on the surface of the vascular endothelial cells hydrolyze these lipoproteins to the cholesterol-rich LDL. These LDL particles have a rather long life and circulate in the blood for about two days before they are cleared. The oxidized form of nLDL, oxLDL, has been shown to exert cytotoxic effects on endothelial cells of the vasculature, leading to endothelial dysfunction [22].

We hypothesized that metformin might be capable of preventing the oxidation of LDL and thereby the development of endothelial dysfunction and atherosclerosis. Our assumption is supported by previous reports having shown that metformin attenuates superoxide generation as well as oxLDL-induced cell death in human primary coronary artery endothelial cells [23].

In the present study, we aimed to investigate the capability of metformin to impede the oxidation of nLDL using a well-established Cu^2+^ in vitro model. nLDL was preincubated in the absence or presence of increasing concentrations of metformin and then oxidized by addition of CuCl_2_. The degree of oxidation of the lipid part of the LDL particle was assessed by measuring lipid hydroperoxide levels (LPO) as well as malondialdehyde (MDA) concentrations. The oxidation of the protein part of the LDL particle was assessed by quantifying oxidation-specific immune epitopes and by determining the respective relative electrophoretic mobility (REM) [24]. Additionally, we aimed to assess the cytotoxic effects of the oxLDL obtained under increasing concentrations of metformin in the vascular endothelial cell line EA.hy926 cells using the alamarBlue viability test [25,26].

The chemical mechanisms by which copper ions oxidize LDL are so far not completely understood. Therefore, elucidation of the precise reactions by which metformin impedes Cu^2+^-mediated LDL oxidation is somewhat difficult. It has been suggested that copper ions form initiating radicals by Fenton-type reactions or by transition complexes with molecular oxygen [20], leading to the formation of hydroxyl radicals (HO^•^), hydroperoxyl radicals (HOO^•^), and superoxide radical anions (O_2_^•−^). By means of quantum chemical calculations, we assessed the reaction capability of these reactive oxygen species (ROS) with metformin in order to reveal the possible LDL oxidation attenuation mechanism.

## 2. Materials and Methods

### 2.1. Preparation of nLDL

The appropriate institutional review board (ethics committee of the Medical University of Graz; 27-320 ex 14/15) approved this study. Written informed consent was obtained from all participants. Human LDL (1.020 to 1.063 g/mL) was obtained from the plasma of four normolipemic (Lp(a) < 5 mg/dL), fasting (12 to 14 h) male young individuals by sequential ultracentrifugation with potassium bromide [27,28]. Pefabloc (50 µM, Sigma Aldrich, Vienna, Austria), EDTA (1 g/L, Merck, Darmstadt, Germany), and butylated hydroxytoluene (BHT, 20 µM, Sigma, St. Louis, MO, USA) were present during all steps of lipoprotein preparation in order to prevent lipid peroxidation and apolipoprotein B (apoB) cleavage by proteinases and possible contaminating bacteria. The samples were sterile filtered and stored at 4 °C in the dark until use. The Lowry method was applied to measure the protein content of LDL [29]. Total cholesterol of the isolated LDL was determined enzymatically using the CHOD-iodide test kit (Boehringer-Mannheim, Mannheim, Germany).

### 2.2. nLDL Oxidation Using Cu^2+^ Ions

nLDL (1.5 mg/mL) was preincubated with increasing concentrations of metformin (0–2500 µg/mL) for 30 min at 37 °C (pH = 7.4) in 0.01 mol/L phosphate buffer containing 0.154 mol/L NaCl. Subsequently, nLDL oxidation was triggered by adding CuCl_2_ (10 µmol/L, final concentration) for up to 8 h as described previously [25,27].

### 2.3. Determination of Lipid Hydroperoxides (LPOs)

The amount of LPO present in LDL of arterial or venous origin was determined with a spectrophotometric assay for lipid hydroperoxides in serum lipoproteins [25,27]. In principle, lipid peroxides are capable of converting iodide to iodine. Briefly, 100 µL of LDL solution (containing 1.5 mg/mL of total LDL) was mixed on a vortex mixer with 1 mL of a color reagent taken from a commercially available kit for enzymatic determination of cholesterol (CHOD-iodide; Merck; Darmstadt, Germany). The samples were allowed to stand for 30 min in the dark at ambient temperature. The absorbance was measured at 365 nm against the color reagent only as the blank, and the concentration was calculated using the molar extinction EM = 2.46 × 104 M^−1^cm^−1^ [25].

### 2.4. Determination of MDA

MDA was determined according to a previously described HPLC method after derivatization with 2,4-dinitrophenylhydrazine (DNPH) [30]. Protein-bound MDA was hydrolyzed and deproteinized as described previously [25]. The supernatant was mixed with 12.5 µL DNPH solution and injected into the HPLC system (injection volume: 40 µL). The MDA standard was prepared as previously described [31,32]. The DNPH derivatives (hydrazones) were isocratically separated, and the utilized HPLC consisted of an L-2200 autosampler, L-2130 HTA pump, and L-2450 diode array detector (all: VWR Hitachi; Vienna; Austria). Detector signals (absorbance at 310 nm) were recorded, and the EZchrom Elite software (VWR) was used for data acquisition and analysis.

### 2.5. Determination of Oxidation-Specific Immune Epitopes

Monoclonal antibodies raised against modified apoB were used to monitor the formation of oxidation-induced epitopes on apoB by means of a solid-phase dissociation-enhanced lanthanide fluorescence immunoassay (DELFIA^®^) as described previously [24]. The anti-ox-apoB (OB 04) was a monoclonal antibody raised against copper-oxidized LDL and characterized to react specifically with oxidized apoB-containing lipoproteins [24]. The anti-apoB was a rabbit polyclonal antibody purchased from Behring (Marburg, Germany). Both were used as detecting antibodies. Eu^3+^-labeled rabbit antimouse IgG (for OB/04) or Eu^3+^-labeled sheep antirabbit IgG (for anti-apoB) were used as reporting antibodies. The microtiter plates (Nunc-MaxiSorpR) were coated with LDL (200 µL at a concentration of 10 µg LDL/mL in a coating buffer containing 1 g/L EDTA) and incubated overnight at 4 °C. The plates were washed after coating three times with washing buffer containing 0.05% Tween-20 using a microplate washer (Gemini, Apeldoorn, The Netherlands). The remaining binding sites were blocked with 250 µL of blocking buffer for 1 h at room temperature. Each plate was washed again as described above. The monoclonal antibody MAB/OB 04 (200 µL/well diluted 1:200 in PBS) against oxidatively modified LDL and a polyclonal rabbit antihuman apoB100 (200 µL/well, 1:10,000 in PBS) were used as detection antibodies. The plates were subsequently incubated at room temperature for 1 h and washed as described above. Then, 200 µL of a goat antimouse IgG (1:1000 in PBS for MAB OB/04 detection) and 200 µL of a goat antirabbit antibody (1:4000 in PBS for antihuman apoB100 detection) were added to each well and incubated at room temperature for 1 h. After incubation, the plates were washed six times. In order to release Eu^3+^, 200 µL of the enhancement solution was added to each well and incubated for five minutes while shaking. The fluorescence in each well was measured using a DELFIA research fluorometer. The amount of oxidation-specific epitopes on the LDL particle was expressed as the ratio of oxidatively modified LDL counts to nLDL counts as described previously [25,30].

### 2.6. Determination of REM

An agarose gel (1%) was used for the electrophoretic runs, and phosphotungstate-Mg^2+^ reagent was utilized to precipitate the lipoproteins on the gel. Electrophoresis was run in 0.05 M barbital buffer at 100 V for 50 min. REM was defined as the ratio of the migration distance of oxLDL and that of nLDL as described previously [25].

### 2.7. Cell Culture

EA.hy926 cells originate from the American Type Culture Collection (ATCC) and were kindly gifted by Dr. C.J.S. Edgell (University of North Carolina, Chapel Hill, NC, USA) [32,33]. EA.hy926 cells were generated by fusing primary human umbilical vein endothelial cells and a thioguanine-resistant clone of the human A549 cell line [32,33]. It has been shown that EA.hy926 cells behave in many aspects like primary endothelial cells [33,34,35]. EA.hy926 cells were cultured as described recently [25]. nLDL as well as oxLDL (prepared by oxidizing nLDL with Cu^2+^ in the presence of increasing concentrations of metformin) were applied to the EA.hy926 cells at a concentration of 0.4 mg/mL for up to 8 h.

### 2.8. Cell Viability (AlamarBlue Assay)

In order to assess the cytotoxic effects of oxLDL on EA.hy926 cells, we used the alamarBlue assay according to the manufacturer’s instructions. AlamarBlue uses the natural reducing power of living cells to convert resazurin to fluorescent resorufin. The amount of fluorescence produced is proportional to the number of living cells and cell viability [26]. Fluorescence was detected with a fluorescence plate reader (POLARstar OPTIMA, BMG Labtech, Offenburg, Germany) with filter set Ex544/Em590 prior to and during incubation (37 °C, 5% CO_2_) as described previously [26].

### 2.9. Quantum Chemical Calculations

Density functional theory (DFT) calculations were conducted using Becke’s three parametric density exchange functional with the correlation function by Lee, Yang, and Parr (restricted for neutral species and unrestricted for radical species). All computations included geometry optimizations and vibrational analyses in order to determine the thermal contributions to free energies. Calculations were performed using the 6–311 + G (2d,p) basis set. The calculations were conducted by simulating either a polar (water) or a nonpolar environment (benzene); here, the SMD model was employed [36]. Changes of Gibb’s free energies (delta G^0^-values) were computed for the reactions considered at standard conditions (298.15 K, 101.325 kPa). For all the molecular structures studied, electron densities, electrostatic potentials, and spin density distributions (in the case of open shell systems) were obtained. All quantum chemical calculations and analyses of the results were conducted employing the GAUSSIAN suite of quantum chemistry programs (GaussView 6.0 and Gaussian G16 W, Gaussian Inc., Pittsburgh, PA, USA).

### 2.10. Statistics

The GraphPad Prism package (v. 8.0) was used for statistical evaluation. One-way ANOVA and Bonferroni post tests were used for statistical evaluation of the effects of increasing concentrations of metformin on indicators of LDL oxidation and on cell viability. Statistical significance was set at *p* ≤ 0.05. * *p* ≤ 0.05, ** *p* ≤ 0.01, *** *p* ≤ 0.001.

## 3. Results

### 3.1. Effect of Metformin on the Oxidizability of the Lipid Part of the LDL Particle

nLDL preparations were oxidized by addition of 10 µmol/L CuCl_2_ for up to 8 h in the absence or presence (100, 250, 500, 1000, and 2500 µg/mL) of metformin. The respective LPO content of LDL, indicating the oxidative status of the lipid part of the LDL particle, time-dependently increased and reached a maximum between four and eight hours of incubation. The time course of LPO formation in the absence of metformin is shown in Figure 1A. Metformin concentration-dependently suppressed the Cu^2+^-triggered oxidation of the lipid part of the LDL particle; formation of both LPOs (Figure 1B) and MDAs (Figure 1C) concentration-dependently decreased in the presence of increasing concentrations of metformin.

### 3.2. Effect of Metformin on the Oxidizability of the Protein Part of the LDL Particle

As stated above, nLDL preparations were incubated with CuCl_2_ for up to 8 h in the absence or presence of metformin. The amount of oxidation-specific epitopes concentration-dependently decreased in the presence of increasing concentrations of metformin (Figure 2A), indicating that metformin is capable of suppressing the oxidation of the protein part of the LDL particle. REM values decreased only slightly, but not significantly, in the presence of increasing concentrations of metformin (Figure 2B).

### 3.3. Cytotoxicity of oxLDL Formed under Increasing Concentrations of Metformin in EA.hy926 Cells

The cytotoxicity of oxLDL in EA.hy926 cells concentration-dependently decreased when it was formed in the presence of increasing concentrations of metformin during Cu^2+^-triggered oxidation of LDL (Figure 3). Cell viability was significantly reduced in oxLDL-incubated cells but was restored when using oxLDLs formed under increasing concentrations of metformin.

### 3.4. Reactions of Metformin with Hydroxyl, Hydroperoxyl, or Superoxide Radical Anion

Gibbs’s free energies for the abstraction of one hydrogen atom from metformin by the hydroxyl (to yield water), hydroperoxyl (to yield hydrogen peroxide), or superoxide (to yield hydroperoxide anion) radical anion were calculated as described above and are shown in Table 1.

The abstraction of each of the five N-centered hydrogens as well as of two methyl-hydrogens were studied according to the scheme shown in Figure 4.

Only the reactions of metformin with HO^•^ were exergonic. This is indicated by the negative values for the respective Gibb’s free energies in both polar (water) and apolar (benzene) milieus, with abstraction of the C-centered hydrogen atom (number 7) resulting in the highest negative energy values. Therefore, HO^•^ is the most suitable candidate for scavenging through metformin. Reactions of both HOO^•^ and O_2_^•−^ with metformin were endergonic (positive values of the respective Gibb’s free energies) and are therefore unlikely to occur spontaneously.

## 4. Discussion

In the present study, we show that the hypoglycemic agent metformin is capable of attenuating the oxidation of nLDL. Oxidized LDL results from modification of LDL by lipidperoxidation and plays a crucial role in the development of endothelial dysfunction and atherosclerosis [37,38].

Using a well-established in vitro model in which nLDL oxidation is accomplished by addition of Cu^2+^ ions [39], we found that metformin concentration-dependently impeded the formation of LPOs and MDA in the LDL particle. Thus, metformin is apparently capable of impeding the oxidation of the lipid moiety of the LDL particle. We also found a concentration-dependent decrease in the number of oxidation-specific immune epitopes on the LDL particle in the presence of increasing concentrations of metformin during oxidation, indicating impeded oxidation of the protein moiety of the LDL particle. Correspondingly, we also found a decrease in oxLDL-induced cytotoxicity in EAhy.926 cells with oxLDL formed under increasing concentrations of metformin.

The antioxidant efficacy of metformin is comparable to that of ethyl pyruvate, as shown in our previous study [25]. For example, the formation of LPOs in the LDL particle was halved in the presence of 500 µg/mL of metformin and ethyl pyruvate.

The chemical events responsible for copper-induced oxidation of nLDL are still somewhat elusive. Lipid hydroperoxides, often present in LDL preparations, become decomposed by Cu^2+^ to alkoxyl and peroxyl radicals, which are believed to initiate oxidation of polyunsaturated fatty acids through abstraction of a hydrogen atom. The oxidation becomes propagated via oxygenation [22]. Additionally, it has been suggested that lipid peroxidation is initiated by the hydroxyl radical (HO^•^), which is formed via sequential reduction of oxygen (via hydrogen peroxide and superoxide) by Cu^+^ [40]. Superoxide (O_2_^•−^) can become partially protonated to yield the hydroperoxyl radical (HOO^•^), which is very stable and, due to its uncharged nature, might diffuse to the core of the LDL particle and initiate the peroxidation of polyunsaturated fatty acids [25,41].

The attenuating effect of metformin on the oxidation of nLDL shown in our study suggests that metformin may react with at least one of the three radicals mentioned above, i.e., HO^•^, HOO^•^, or O_2_^•−^. Our calculations of reaction free energies identified the hydroxyl radical as the most likely candidate to be scavenged by metformin. Therefore, we propose that the capability of metformin to impede oxidation of nLDL is attributable, at least partially, to its capability to scavenge hydroxyl radicals.

The results of our calculations are supported by several studies dealing with the capability of metformin to react with ROS. Bonnefont-Rousselot et al. have shown that metformin is able to scavenge hydroxyl radicals but not superoxide radicals and hydrogen peroxide [42,43]. Collin et al. have shown that metformin reacts with hydroxyl radicals under the formation of four oxidation end-products, e.g., the hydroperoxide of metformin or a covalent dimer of metformin [44].

The major cause of premature death in diabetes is cardiovascular disease (CVD), and arteriosclerosis in turn is a precondition for the development of CVD [45,46]. Prolonged residence times of highly atherogenic oxLDL is a key feature of diabetic CVD [47], and numerous studies have shown that oxidation of LDL particularly renders this particle atherogenic [37].

Apparently, drugs that are capable of attenuating the oxidation of LDL possess antiatherogenic properties and could therefore prevent the development of CVD [48]. Metformin is, according to the findings of our present study, such a drug. Numerous observational studies confirm this assumption. Strong beneficial effects of metformin therapy concerning combined mortality and cardiovascular events [49], reduced risk for all-cause mortality for diabetic patients under metformin treatment compared to nondiabetic individuals [50], and reduction in cardiovascular mortality as well as incidence of cardiovascular events under metformin treatment have been shown [51,52]. However, most of these studies focused on the correlation between metformin intake and decreased atherosclerotic/vascular events in patients with diabetes without providing any underlying molecular mechanisms.

Our present study provides one indication for the reduced prevalence of atherosclerosis/CVD in patients with diabetes under metformin therapy: the prevention/attenuation of oxidation of nLDL via inactivation of hydroxyl radicals by metformin. Consistent therewith, a clinical study has shown decreased oxidative damage in the apoB, the protein moiety of the LDL particle [45,53].

A limitation of our study is that the antioxidant action of metformin might be attributable to its capability to not only scavenge hydroxyl radicals but to also bind copper, thereby lowering effective Cu^2+^ concentrations during the nLDL oxidation procedure [54]. However, efficient oxidation of nLDL occurred even in the presence of a large surplus of metformin, indicating that this complex formation is of minor importance under our experimental setting.

Besides impeding the oxidation of nLDL, shown in our present study, several further modes of action have been reported rendering metformin an antiatherosclerotic drug, namely, the regulation of oxLDL-provoked endothelial dysfunction through upregulation of sirtuin 1 expression [55], protection of the vasculature by activation of endothelial nitric oxide synthase [56], and decreased inflammatory activity in patients taking metformin [57]. Furthermore, it has been suggested that the vasoprotective effects of metformin are attributable to its capability to impede mitochondrial fission in endothelial cells [58], its antihypertensive effects [59], its capability to improve dyslipidemia [60], and its capability to reduce the uptake of lipids by macrophages with subsequent apoptosis, all key steps in atherogenesis [61,62]. Moreover, it has been shown that metformin attenuates myocardial ischemia–reperfusion injury via the upregulation of antioxidant enzymes and via involvement of adenosine-monophosphate-activated protein kinase (AMPK) [63]. In addition, a combined therapy of metformin with the new agents SGLT2 inhibitors or GLP1 receptor agonists has more recently been suggested to be advantageous for patients with high cardiovascular risk [39].

The unique properties of metformin extend its range of action beyond cardiovascular protection to age-related diseases, such as cancer and dementia. A decrease in cancer incidence, an increase in cancer-specific survival, and a decrease in the incidence of dementia in people with type 2 diabetes under metformin therapy have been shown in recent meta-analyses [64,65,66].

## 5. Conclusions

In summary, our study adds a further mechanism through which metformin acts as an antiatherogenic agent: its capability to impede oxidation of LDL through scavenging hydroxyl radicals. Metformin currently plays an important role in the treatment of type 2 diabetes. Our results suggest that metformin could also be used to treat further diseases that are associated with inflammation/oxidative stress in the vasculature, both in diabetic and nondiabetic populations [3,67].

## Figures and Tables

**Figure 1 pharmaceutics-15-02111-f001:**
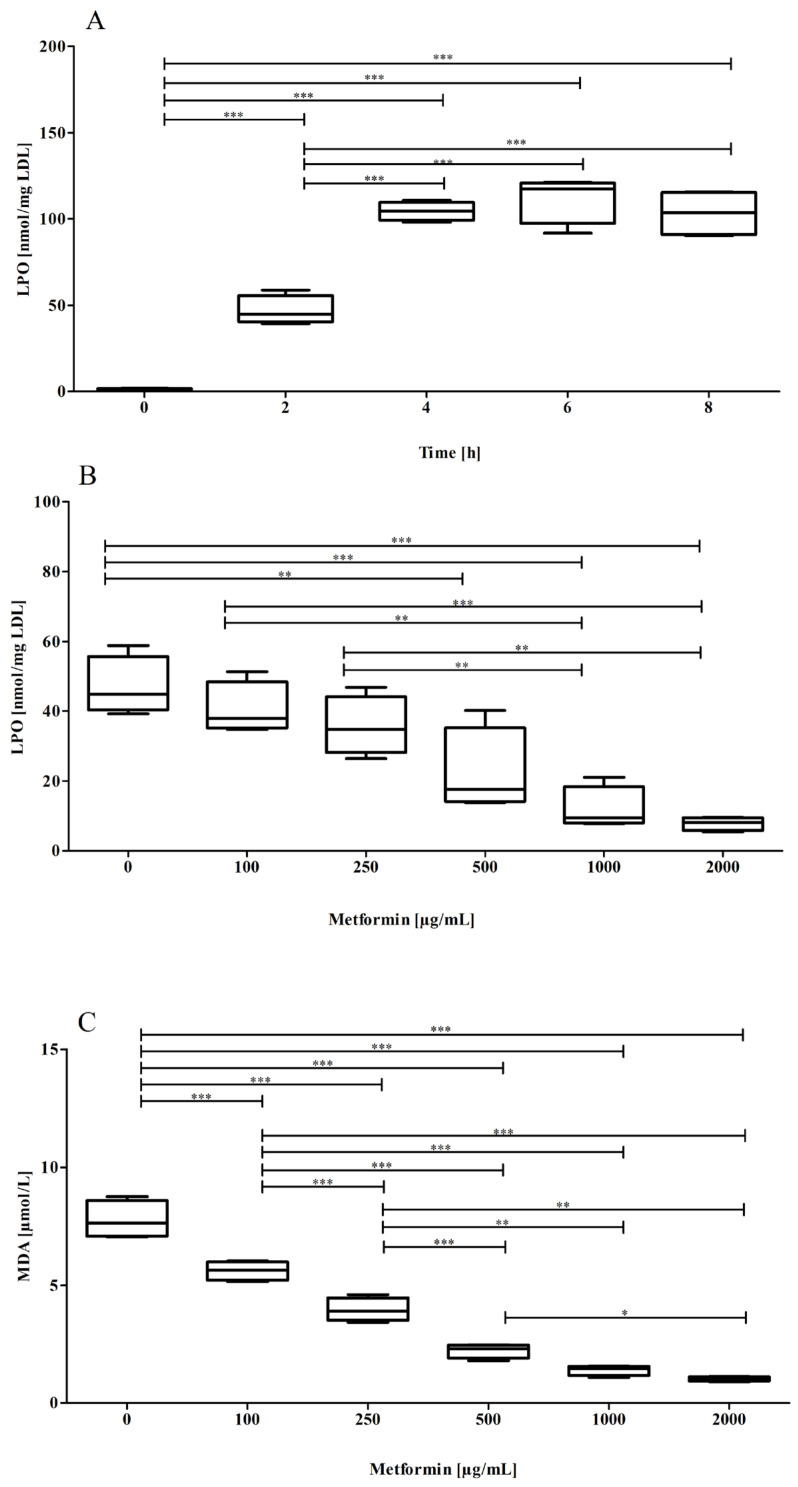
Cu^2+^-induced oxidation of the lipid part of the LDL particle. The LPO content of LDL significantly increased in the course of Cu^2+^-triggered oxidation (*p* < 0.0001, ANOVA) (**A**). Metformin significantly (*p* < 0.0001, ANOVA) suppressed LPO (**B**) and MDA (**C**) formation after 2 h of incubation time. Data represent mean ± SD from four separate measurements. * *p* ≤ 0.05, ** *p* ≤ 0.01, *** *p* ≤ 0.001.

**Figure 2 pharmaceutics-15-02111-f002:**
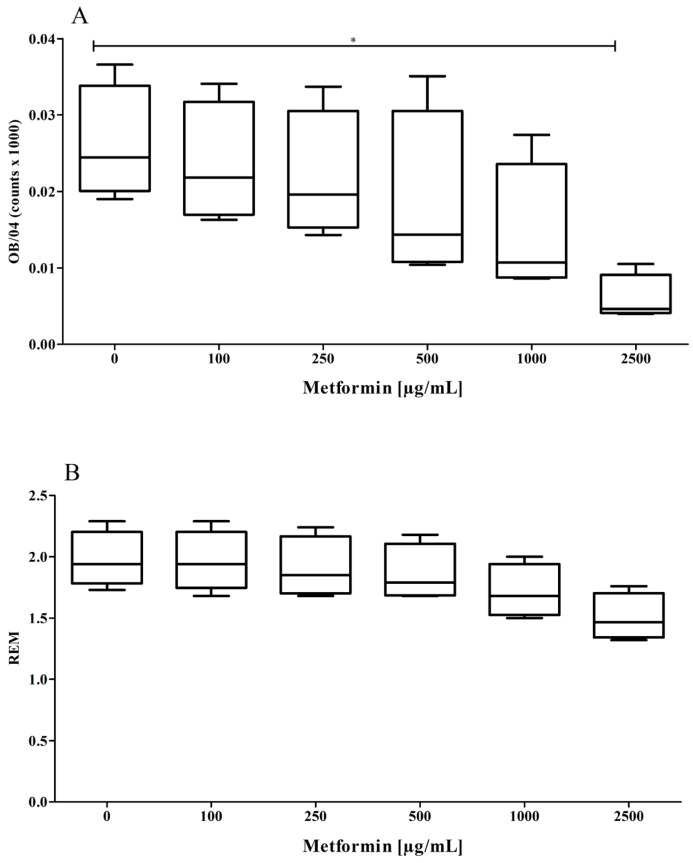
Cu^2+^-induced oxidation of the protein part of the LDL particle. Metformin significantly (*p* = 0.0031, ANOVA) suppressed the formation of oxidation-specific epitopes on the LDL particle (**A**) after 6 h incubation time. REM values were numerically decreased albeit not reaching statistical significance ((**B**), *p* = 0.069). Data represent mean ± SD from four separate measurements. * *p* ≤ 0.05.

**Figure 3 pharmaceutics-15-02111-f003:**
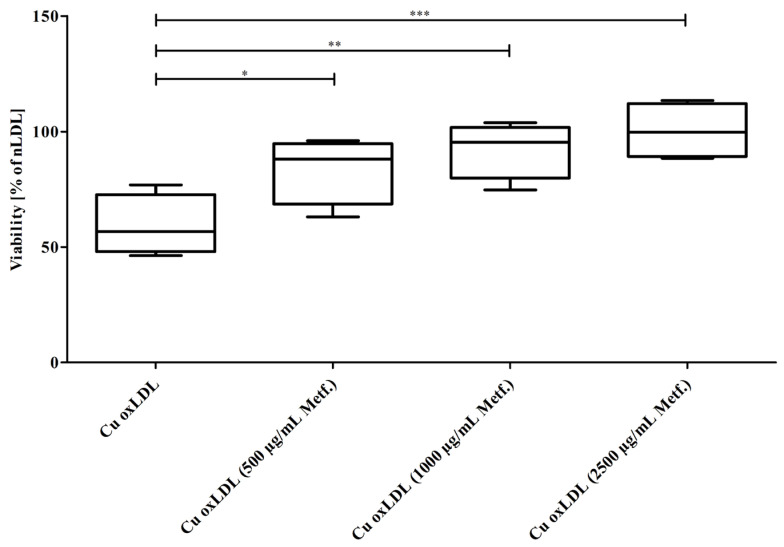
Metformin during LDL oxidation mitigated the cytotoxic effects of oxLDL. nLDL was preincubated with 0, 500, 1000, or 2500 µg/mL of metformin and then oxidized with CuCl_2_ until LPO levels reached 100 nmol/mg of LDL protein. Subsequently, the EAhy.926 cells were incubated with the thus obtained oxLDLs (0.4 mg/mL). Cell viability concentration-dependently increased with oxLDLs formed under increasing levels of metformin. Data represent mean ± SD of four separate measurements. * *p* ≤ 0.05, ** *p* ≤ 0.01, *** *p* ≤ 0.001.

**Figure 4 pharmaceutics-15-02111-f004:**
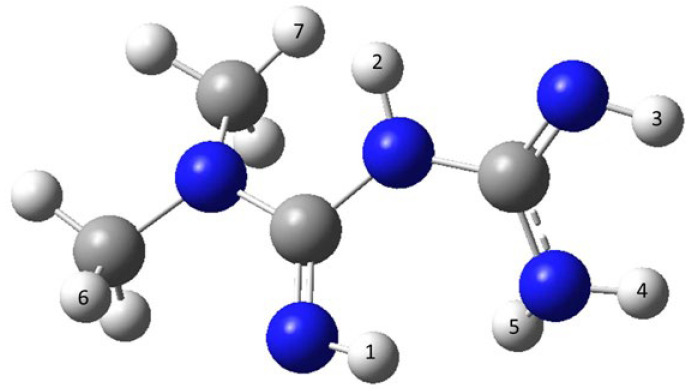
Chemical structure of metformin. Large dark spheres: nitrogen; large light spheres: carbon; small spheres: hydrogen.

**Table 1 pharmaceutics-15-02111-t001:** Gibb’s free energies of reaction. Negative values of Gibb’s free energy mark exergonic reactions (abstraction of one hydrogen atom from metformin). Positive values mark endergonic reactions.

Radical	Position of H-Abstraction	Gibbs’ Free Energy (kcal/mol)	
		Water	Benzene
	1	−25.25	−23.95
	2	−25.19	−20.78
Hydroxyl radical	3	−21.89	−19.05
	4	−20.37	−18.43
	5	−20.51	−18.34
	6	−27.35	−26.60
	7	−28.55	−28.07
	1	24.01	31.22
	2	24.07	34.38
Superoxide radical anion	3	27.37	36.11
	4	28.89	36.73
	5	28.74	36.83
	6	21.91	28.57
	7	20.70	27.10
	1	8.24	9.53
	2	8.30	12.70
Hydroperoxyl radical	3	11.60	14.42
	4	13.12	15.04
	5	12.98	15.14
	6	6.14	6.88
	7	4.94	5.41

## Data Availability

The authors hereby declare that the data presented in this study will be presented upon request by the corresponding author.

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
