# Peer review of "Metformin Impedes Oxidation of LDL In Vitro"

_pharmaceutics, 2023, doi:10.3390/pharmaceutics15082111_

Round 1
Reviewer 1 Report
What is metformin and what is it generally used for?
What are LDLs, and why is it important to prevent their oxidation?
What is LDL oxidation, and what is its connection to cardiovascular diseases?
How was the study conducted to determine if metformin prevents LDL oxidation in vitro?
What were the results of the study on the effect of metformin on LDL oxidation in vitro?
What are the implications of these findings for cardiovascular health?
Are there other studies or research that have examined the effect of metformin on LDLs and cardiovascular health?
What are the potential mechanisms by which metformin could prevent LDL oxidation?
Are there any limitations or precautions to consider regarding these results?
Besides preventing LDL oxidation, does metformin have other beneficial effects on health?
Author Response
Comments to Reviewer 1:
We thank the reviewer for the valuable comments allowing improvement of our manuscript.
What is metformin and what is it generally used for?
Metformin is a guanidine derivative and was initially extracted from the plant Galega officinalis (French lilac). For more than 60 years it has been used as a glucose-lowering medication in humans. Metformin is highly cost-effective and has proven to be safe. Metformin is considered as insulin-sensitizer by virtue of its capability to reduce insulin resistance and to decrease plasma fasting insulin levels.
This is stated now in lines 43-47.
What are LDLs, and why is it important to prevent their oxidation?
The liver assembles triglyceride-rich lipoproteins and secretes them into the circulation. Lipoprotein lipases on the surface of the vascular endothelial cells hydrolyze these lipoproteins to the cholesterol-rich LDL. These LDL particles have a rather long life and circulate in the blood for about two days before they are cleared. The oxidized form of nLDL, oxLDL, has been shown to exert cytotoxic effects on endothelial cells of the vasculature, leading to endothelial dysfunction.
This is stated now in lines 94-68.
What is LDL oxidation, and what is its connection to cardiovascular diseases?
Low-density lipoprotein (LDL), not atherogenic in its native state (nLDL), might become subject to oxidation in the subendothelial space of arteries by mechanisms involving free radicals and/or lipoxygenases. LDL lipids undergo peroxidation. Subsequently, certain aldehydes generated during lipid peroxidation modify the apolipoprotein B100 (apoB100), the protein part of LDL. The resulting oxidized form of LDL (oxLDL) is readily internalized by macrophages through a so-called “scavenger receptor” pathway. The macrophages change to foam cells, initiating processes resulting in atherosclerosis.
This is stated now in lines 73-75 and 90-94.
How was the study conducted to determine if metformin prevents LDL oxidation in vitro?
We aimed to investigate whether metformin is capable to impede the oxidation of LDL by using a well-established Cu2+ in vitro model. nLDL was preincubated in the absence or presence of increasing concentrations of metformin and then oxidized by addition of CuCl2. The degree of oxidation of the lipid part of the LDL particle was assessed by measuring lipid hydroperoxide levels (LPO) as well as malondialdehyde (MDA) concentrations. The oxidation of the protein part of the LDL particle was assessed by quantifying oxidation-specific immune epitopes and by determining the respective relative electrophoretic mobility (REM).
This is stated in lines 106-113.
What were the results of the study on the effect of metformin on LDL oxidation in vitro?
We found that metformin concentration-dependently impeded the formation of lipid hydroperoxides, malondialdehyde, and oxidation-specific immune epitopes when oxidation of LDL was initiated by addition of Cu2+. The cytotoxicity of oxLDL in EA.hy926 vascular endothelial cells was reduced when it was obtained under increasing concentrations of metformin.
What are the implications of these findings for cardiovascular health?
We believe that the main finding of our study is that metformin is not only an efficient drug for lowering blood glucose but also capable of impeding the oxidation of LDL, an early and important step in the development of atherosclerosis, and, subsequently, cardiovascular complications. Up to 75% of patients with type 2 diabetes die from cardiovascular complications.
Are there other studies or research that have examined the effect of metformin on LDLs and cardiovascular health?
Yes, there are studies (cited in the manuscript, 50-52) reporting that metformin therapy has beneficial effects on the mortality of atherosclerosis/cardiovascular events, however without providing any underlying mechanisms. Our study provides one indication for the reduced prevalence of atherosclerosis/CVD in patients with diabetes under metformin therapy: the prevention/attenuation of oxidation of nLDL via inactivation of hydroxyl radicals by metformin.
Rabbani et al. have shown that metformin is capable of decreasing the oxidative damage in the apoB, the protein moiety of the LDL particle. This study is cited in our manuscript as [46]. The major oxidative marker was methionine sulfoxide residues with mean content of 2084 pmol/mg apoB100, equivalent to 1.07 mol/mol apoB100 or 13.7 mmol/mol Met.
What are the potential mechanisms by which metformin could prevent LDL oxidation?
Elucidation of the mechanisms underlying metformin-induced attenuation of LDL oxidation is difficult since the chemical mechanisms by which copper ions oxidize LDL in vitro are still somewhat elusive.
Therefore, we have calculated Gibbs’s free energies for the abstraction of one hydrogen atom from metformin by the hydroxyl radical (to yield water), by the hydroperoxyl radical (to yield hydrogen peroxide), or by the superoxide radical anion (to yield hydroperoxide anion). Only the reactions of metformin with HO• were exergonic. We propose that the capability of metformin to impede oxidation of nLDL is attributable, at least partially, to its capability to scavenge hydroxyl radicals.
Are there any limitations or precautions to consider regarding these results?
The drawback of our in vitro experiments is utilization of an isolated chemical system. It was not feasible to precisely imitate the natural environment of the subendothelial space of arteries.
Besides preventing LDL oxidation, does metformin have other beneficial effects on health?
Yes, several beneficial effects of metformin therapy have been reported. Metformin is capable of protecting the vasculature by activation of endothelial nitric oxide synthase and to decrease inflammatory activity in patients taking metformin. Furthermore, metformin has vasoprotective effects by impeding mitochondrial fission in endothelial cells. Metformin has anti-hypertensive effects and can improve dyslipidemia. Furthermore, metformin can reduce the uptake of lipids by macrophages with subsequent apoptosis. Metformin attenuates myocardial ischemia-reperfusion injury via up-regulation of antioxidant enzymes and via involvement of adenosine monophosphate-activated protein kinase (AMPK).
This is stated in lines 412-426.
Reviewer 2 Report
The paper entitled “Metformin impedes oxidation of LDL in vitro” includes potentially relevant scientific data for the pharmacological issues of potential implementation in management of diabetes mellitus, obesity as well as metabolic syndrome. Authors of the publication in question – among others – tend to suggest that “…Metformin, beside its glucose-lowering effect, might be a suitable agent to impede the development of atherosclerosis and associated CVD. This is due to its capability to impede LDL oxidation, most likely by scavenging hydroxyl radicals…”.
However, I have a few remarks:
1. The sentence "...Nowadays, metformin is the recommended first-line treatment for type 2 diabetes [5,6]..." may confuse the reader - why?
It's true that until a few years ago this was the case, for example, in 2017.
Now we are in 2023 and the ADA standards indicate that metformin can be the active pharmaceutical substance used in the initial pharmacological approach - but not in all cases. The quoted passage should be modified after a careful reading of the ADA recommendations for 2023
9 Pharmacologic Approaches to Glycemic Treatment:
Standards of Care in Diabetes-2023
Diabetes Care 2023;46(Suppl. 1):S140-S157 | https://doi.org/10.2337/dc23-S009
2. In the introduction, the Authors describe only the so-called advantages of metformin - unfortunately, like any active pharmaceutical substance - metformin is also characterized by side effects and contraindications, which must be briefly but nevertheless cited.
3. the literature on the basis of which the introductory chapter was developed is outdated - progress in medical and pharmaceutical science is made every day - the literature must be updated.
4. "Materials and Methods" subsections (2.2 nLDL oxidation using Cu2+ ions; 2.6 Determination of REM; 2.8 Cell viability (alamarBlue assay)) do not have literature references - must be supplemented.
5. The use of selected statistical methods should be convincingly justified in order to avoid doubts about the reliable control of the results.
6. Figures 1,2,3 have insufficient resolution should be corrected.
7. Selected brief but relevant parts of the “Discussion” were not referred to the results of other authors (lines: 297-303; 318-323.).
8. Conclusions, although answering the scientific question included in the study aim, however, are laconic and vague - must be modified.
Author Response
Comments to Reviewer 2:
We thank the reviewer for the valuable comments allowing improvement of our manuscript.
- The sentence "...Nowadays, metformin is the recommended first-line treatment for type 2 diabetes [5,6]..." may confuse the reader - why?
It's true that until a few years ago this was the case, for example, in 2017.
Now we are in 2023 and the ADA standards indicate that metformin can be the active pharmaceutical substance used in the initial pharmacological approach - but not in all cases. The quoted passage should be modified after a careful reading of the ADA recommendations for 2023
9 Pharmacologic Approaches to Glycemic Treatment:
Standards of Care in Diabetes-2023
Diabetes Care 2023;46(Suppl. 1):S140-S157 | https://doi.org/10.2337/dc23-S009
As suggested by the reviewer, the passage “Nowadays, metformin is the recommended first-line treatment for type 2 diabetes” is changed now to:
“Metformin plays an important role in pharmacologic therapy for adults with type 2 diabetes. The American Diabetes Association (ADA) recommends that metformin should be continued upon initiation of insulin therapy (unless contraindicated or not tolerated) for ongoing glycemic and metabolic benefit (ElSayed). Metformin is effective, safe, inexpensive, and available in an immediate-release form for twice-daily dosing or as an extended-release form that can be given once daily. Because type 2 diabetes is a progressive disease, combination therapy might be required. Stepwise addition of medications to metformin is recommended to maintain glucose levels at target. More extensive initial combination therapy for more rapid attainment of glycemic goals and later combination therapy for longer durability of glycemic effect is required.”
This is stated now in lines 50-59.
- In the introduction, the Authors describe only the so-called advantages of metformin - unfortunately, like any active pharmaceutical substance - metformin is also characterized by side effects and contraindications, which must be briefly but nevertheless cited.
As suggested by the reviewer, we describe now in the “Introduction section” possible side effects and contraindications of metformin use:
“The principal side effects of metformin use are gastrointestinal intolerance due to bloating, abdominal discomfort, diarrhea, and lactic acidosis. Metformin may be safely used in people with reduced estimated glomerular filtration rates [5]. Moreover, metformin use could be associated with vitamin B12 deficiency and worsening of neuropathy [9]. These side effects occur very rarely.”
This is stated now in lines 60-64.
- the literature on the basis of which the introductory chapter was developed is outdated - progress in medical and pharmaceutical science is made every day - the literature must be updated.
We have added now current literature to the introductory chapter:
ElSayed, N.A.; Aleppo, G.; Aroda, V.R.; Bannuru, R.R.; Brown, F.M.; Bruemmer, D.; Collins, B.S.; Hilliard, M.E.; Isaacs, D.; Johnson, E.L.; et al. 9. Pharmacologic Approaches to Glycemic Treatment: Standards of Care in Diabetes-2023. Diabetes Care 2023, 46, S140-S157, doi:10.2337/dc23-S009.
Aroda, V.R.; Gonzalez-Galvez, G.; Gron, R.; Halladin, N.; Haluzik, M.; Jermendy, G.; Kok, A.; Orsy, P.; Sabbah, M.; Sesti, G.; et al. Durability of insulin degludec plus liraglutide versus insulin glargine U100 as initial injectable therapy in type 2 diabetes (DUAL VIII): a multicentre, open-label, phase 3b, randomised controlled trial. Lancet Diabetes Endocrinol 2019, 7, 596-605, doi:10.1016/S2213-8587(19)30184-6.
Dahl, D.; Onishi, Y.; Norwood, P.; Huh, R.; Bray, R.; Patel, H.; Rodriguez, A. Effect of Subcutaneous Tirzepatide vs Placebo Added to Titrated Insulin Glargine on Glycemic Control in Patients With Type 2 Diabetes: The SURPASS-5 Randomized Clinical Trial. JAMA 2022, 327, 534-545, doi:10.1001/jama.2022.0078.
Out, M.; Kooy, A.; Lehert, P.; Schalkwijk, C.A.; Stehouwer, C.D.A. Long-term treatment with metformin in type 2 diabetes and methylmalonic acid: Post hoc analysis of a randomized controlled 4.3year trial. J Diabetes Complications 2018, 32, 171-178, doi:10.1016/j.jdiacomp.2017.11.001.
Rey-Renones, C.; Baena-Diez, J.M.; Aguilar-Palacio, I.; Miquel, C.; Grau, M. Type 2 Diabetes Mellitus and Cancer: Epidemiology, Physiopathology and Prevention. Biomedicines 2021, 9, doi:10.3390/biomedicines9101429
Zhou, J.B.; Tang, X.; Han, M.; Yang, J.; Simo, R. Impact of antidiabetic agents on dementia risk: A Bayesian network meta-analysis. Metabolism 2020, 109, 154265, doi:10.1016/j.metabol.2020.154265.
Samaras, K.; Makkar, S.; Crawford, J.D.; Kochan, N.A.; Wen, W.; Draper, B.; Trollor, J.N.; Brodaty, H.; Sachdev, P.S. Metformin Use Is Associated With Slowed Cognitive Decline and Reduced Incident Dementia in Older Adults With Type 2 Diabetes: The Sydney Memory and Ageing Study. Diabetes Care 2020, 43, 2691-2701, doi:10.2337/dc20-0892.
- "Materials and Methods" subsections (2.2 nLDL oxidation using Cu2+ ions; 2.6 Determination of REM; 2.8 Cell viability (alamarBlue assay)) do not have literature references - must be supplemented.
As suggested by the reviewer, literature references are stated now:
2.2 nLDL oxidation using Cu2+ ions
Rossmann, C.; Nusshold, C.; Paar, M.; Ledinski, G.; Tafeit, E.; Koestenberger, M.; Bernhart, E.M.; Sattler, W.; Cvirn, G.; Hallstrom, S. Ethyl pyruvate inhibits oxidation of LDL in vitro and attenuates oxLDL toxicity in EA.hy926 cells. PLoS One 2018, 13, e0191477, doi:10.1371/journal.pone.0191477.
el-Saadani, M.; Esterbauer, H.; el-Sayed, M.; Goher, M.; Nassar, A.Y.; Jurgens, G. A spectrophotometric assay for lipid peroxides in serum lipoproteins using a commercially available reagent. J Lipid Res 1989, 30, 627-630.
2.6 Determination of REM
Rossmann, C.; Nusshold, C.; Paar, M.; Ledinski, G.; Tafeit, E.; Koestenberger, M.; Bernhart, E.M.; Sattler, W.; Cvirn, G.; Hallstrom, S. Ethyl pyruvate inhibits oxidation of LDL in vitro and attenuates oxLDL toxicity in EA.hy926 cells. PLoS One 2018, 13, e0191477, doi:10.1371/journal.pone.0191477.
2.8 Cell viability (alamarBlue assay)
Hamalainen-Laanaya, H.K.; Orloff, M.S. Analysis of cell viability using time-dependent increase in fluorescence intensity. Anal Biochem 2012, 429, 32-38, doi:10.1016/j.ab.2012.07.006.
- The use of selected statistical methods should be convincingly justified in order to avoid doubts about the reliable control of the results.
One-way ANOVA and Bonferroni post tests are the standard methods of choice for statistical evaluation of the effects of increasing concentrations of antioxidants on indicators of LDL oxidation and on cell viability, see: Rossmann, C.; Nusshold, C.; Paar, M.; Ledinski, G.; Tafeit, E.; Koestenberger, M.; Bernhart, E.M.; Sattler, W.; Cvirn, G.; Hallstrom, S. Ethyl pyruvate inhibits oxidation of LDL in vitro and attenuates oxLDL toxicity in EA.hy926 cells. PLoS One 2018, 13, e0191477, doi:10.1371/journal.pone.0191477.
- Figures 1,2,3 have insufficient resolution should be corrected.
Figures 1,2, and 3 were sent as TIF with 600 dpi resolution the journal via Email.
- Selected brief but relevant parts of the “Discussion” were not referred to the results of other authors (lines: 297-303; 318-323.).
The data concerning the effects of metformin on the lipid and the protein part of the LDL particle are scarce.
Rabbani et al. have shown that metformin is capable of decreasing the oxidative damage in the apoB, the protein moiety of the LDL particle. This study is cited in our manuscript as [46]. The major oxidative marker was methionine sulfoxide residues with mean content of 2084 pmol/mg apoB100, equivalent to 1.07 mol/mol apoB100 or 13.7 mmol/mol Met.
Bonnefont-Rousselot et al. have shown that metformin is able to scavenge hydroxyl radicals but not superoxide radicals and hydrogen peroxide [43]. Collin et al. have shown that metformin reacts with hydroxyl radicals under the formation of four oxidation end-products, e.g., the hydroperoxide of metformin or a covalent dimer of metformin [45].
- Conclusions, although answering the scientific question included in the study aim, however, are laconic and vague - must be modified.
I ask the reviewer for understanding. I am a chemist and do not dare to apply these results directly to everyday clinical practice in the treatment of diabetics. I can only reflect in the conclusions section the main message of our paper, namely that metformin is able to prevent the oxidation of low-density lipoprotein and is therefore probably also able to attenuate the development of arteriosclerosis and related diseases.
Therefore, the conclusions section has been changed to (lines 437-442) :
“In summary, our study adds a further mechanism through which metformin acts as an anti-atherogenic agent: its capability to impede oxidation of LDL through scavenging hydroxyl radicals. Metformin currently plays an important role in the treatment of type 2 diabetes. Our results suggest that metformin could also be used to treat further diseases that are associated with inflammation/oxidative stress in the vasculature, both in the diabetic as well as in the non-diabetic population [3,68].”
Round 2
Reviewer 2 Report
The Authors introduced all the necessary and appropriate modifications to the text of the article in question.
In its current form, the Article is suitable for publication without any corrections.